# Mitral Valve Systolic Anterior Motion in Robotic Thoracic Surgery as the Cause of Unexplained Hemodynamic Shock: From a Case Report to Recommendations

**DOI:** 10.3390/jcm11206044

**Published:** 2022-10-13

**Authors:** Fabrizio Monaco, Filippo D’Amico, Gaia Barucco, Margherita Licheri, Pierluigi Novellis, Paola Ciriaco, Giulia Veronesi

**Affiliations:** 1Department of Cardiothoracic and Vascular Anesthesia, San Raffaele Scientific Institute, 20132 Milan, Italy; 2Department of Thoracic Surgery, San Raffaele Scientific Institute, 20132 Milan, Italy; 3School of Medicine, Vita-Salute San Raffaele University, 20132 Milan, Italy

**Keywords:** systolic anterior motion, left ventricular outflow obstruction, lung surgery, robotic surgery, hemodynamic instability, shock

## Abstract

Robotic major lung resection for lung cancer carries a risk for intraoperative hemodynamic instability. Systolic anterior motion (SAM) of the mitral valve is a rare and often misrecognized cause of intraoperative hemodynamic instability. If not promptly recognized, SAM leads to a complicated perioperative course. Here, we report for the first time a case of a patient with SAM with a severe degree of left ventricular outflow obstruction (LVOTO) undergoing robotic lung lobectomy and its challenging intraoperative management. A 70-year-old man undergoing robotic left upper lobectomy developed immediately after the induction of general anesthesia hemodynamic instability due to SAM-related LVOTO. The diagnosis was possible, thanks to the use of transesophageal echocardiography (TEE). The treatment strategies applied were preload optimization without fluid overload, ultra-short-acting beta-blockers, and vasopressors. Peripheral nerve blockades were preferred over epidural analgesia to avoid vasodilatation. The patient reported a good quality of recovery and no pain the day after surgery. The management of patients with higher risk of SAM and LVOTO development during robotic thoracic surgery requires a dedicated and skilled team together with high-impact treatment strategies driven by TEE. Since current guidelines do not recommend the use of TEE, even for patients with higher cardiac risk undergoing noncardiac surgery, the present case report may stimulate interest in future recommendations.

## 1. Introduction

Systolic anterior motion (SAM) describes the dynamic movement of the mitral valve (MV) anterior leaflet during systole towards the left ventricular outflow tract (LVOT). SAM could lead to MV regurgitation, LVOT obstruction (LVOTO), and consequent hemodynamic instability, potentially followed by hemodynamic instability. Clinical manifestations are hypotension, tachycardia, pulmonary hypertension, high wedge pressure, and in the worst scenario, pulmonary edema and shock unresponsive to inotropic support. Although SAM usually occurs in patients with hypertrophic cardiomyopathy (HCM) and after cardiac surgery for MV repair [1], this complication may be observed in perioperative settings even in patients without underlying heart diseases during hypovolemia and bleeding [2]. Since there are several predisposing conditions associated with perioperative SAM and LVOTO development in patients who not show SAM at rest, these complications remain not fully predictable. In fact, in patients with predisposing underlying conditions, LVOTO and SAM require provocative maneuvers (i.e., Valsalva maneuver) to become clinically evident. For this reason, the use of transesophageal echocardiography (TEE) should be strongly recommended in the event of unexplained hemodynamic instability in noncardiac surgery. Indeed, intraprocedural TEE allows both early detection and assessment of the degree of hemodynamic impairment, and it has the potential to influence clinical and therapeutic decision making for cardiac surgical patients. Since current guidelines do not recommend the use of TEE in patients undergoing noncardiac surgery (III C) [3], it may be assumed that cases of SAM in noncardiac surgery are largely under-reported, and possible complications such as sudden shock are poorly managed. Pulmonary artery catheter and central line measurements in patients who develop refractory hypotension unresponsive to conventional therapies mimic values found in cardiogenic shock (e.g., low stroke volume, high central pressure, high wedge pressure, pulmonary hypertension), which is usually treated with inotropes and diuretics. Indeed, administration of inotropes and diuretics further worsens SAM and the degree of LVOTO. Generally, patients undergoing thoracic surgery have a greater risk of SAM, regardless of the kind of approach, than other noncardiac procedures as one lung ventilation (OLV) is usually required. Indeed, OLV may be associated with higher intrathoracic pressure and relative hypovolemia, which are both well-known risk factors for SAM and LVOTO [4,5]. Notably, this risk is even higher for robotic-assisted thoracic surgery (RATS) compared with open and video-assisted thoracoscopic surgery (VATS) due to the combination of carbon dioxide (CO_2_) insufflation and the patient’s position. In fact, in contrast with VATS in which the intrapleural space is open to the atmospheric pressure, RATS often needs CO_2_, insufflated through sealed ports, to maximize the intrapleural space by lung deflation and diaphragm flattening [6]. Hence, CO_2_ insufflation, increasing the intrathoracic pressure, may cause both hypotension due to pressure-induced compression of the mediastinal vessels and gas exchange imbalance due to CO_2_ embolization or suboptimal pulmonary ventilation [7,8]. Additionally the use of a more pronounced reversed Trendelenburg position required in RATS may further decrease the venous return eliciting SAM and LVOTO [9]. All these factors, acting together, decrease preload and afterload and may induce tachycardia. Thus, those who work in thoracic surgery should consider SAM and LVOTO as relatively rare causes of reversible shock in patients with hypotension refractory to fluid administration. We report a case of a patient undergoing robotic lung lobectomy who developed SAM and its complicated intraoperative management. Since the diagnosis of SAM deeply affects ventilation, hemodynamics, and pain management, even more in RATS than in noncardiac surgeries, specific considerations are discussed. 

## 2. Case Report

A 70-year-old man (body weight, 90 kg; height, 173 cm) was referred to the thoracic surgery department to undergo robotic left upper lobectomy for stage IIA squamous cell carcinoma. On arrival, the vital signs were: respiratory rate, 17 breaths/min; temperature, 36.4 °C; blood pressure, 110/60 mmHg; heart rate, 45 beats/min; oxygen saturation level, 99% at room air; Glasgow Coma Scale score, 15. A 12-lead electrocardiogram showed sinus bradycardia, right bundle branch block, and left ventricular hypertrophy signs. His past medical history included chronic kidney disease stage IV, hypertension, hypertrophic ischemic cardiomyopathy NYHA class I treated with angioplasty in 2019, alcoholic liver disease with thrombocytopenia, metabolic syndrome, peripheral vascular disease, depressive disorder, and benign prostatic hyperplasia. Echocardiographic findings revealed severe concentric hypertrophy of LV more localized to the apical region and septum. The basal IVS thickness was 19 mm. The ejection fraction was 70% with no regional motion abnormalities and no significant valvular defects. Surgical intervention was performed the day after the admission. A low dose of propofol (1 mg/kg) was used for induction, and sevoflurane was used to maintain anesthesia. During induction, the pressure dropped to 50 mmHg, and desaturation occurred. This complication was managed with fluids and pure alpha-agonist administration along with a reduction of the volume of insufflation during manual ventilation to decrease intrathoracic pressure. After induction, paravertebral block with ropivacaine 7.5% 20 mL was performed with the patient positioned in lateral position to allow opioid-free anesthesia and postoperative analgesia. In the operating room, the patient was monitored with TEE throughout the procedure. During the surgery, TEE revealed a gradient of obstruction ranging between 60 and 180 mmHg (Figure 1, Appendix A, Appendix A). Preload fluctuations associated with robotic surgery progressively led to persistent refractory hypotension (MAP < 50 mmHg) unresponsive to fluid administration. Interestingly, a new onset of right ventricle dysfunction was observed as the effect of the increased wedge pressure and increased systolic pulmonary pressure. Thus, to avoid fluid overload, which is harmful in advanced renal failure (RIFLE stage IV [10]), and to manage refractory hypotension, a continuous infusion of norepinephrine 0.1 mcg/kg/min was started. The patient was extubated immediately after the end of the surgical procedure. He was discharged to the ward after 3 hours of observation in the recovery room. Additionally, interscalene block (ropivacaine 2% 10 mL) was performed to control the onset of ipsilateral shoulder pain. Norepinephrine was progressively weaned under transthoracic monitoring in the recovery room. The quality of recovery and pain control were assessed respectively through the postoperative quality of the recovery score (QoR-15) and numeric pain rating scale (NRS). The value was 130 for QoR-15 and 2 for NRS 24 h after surgery. The patient was discharged from the hospital on postoperative day 7, with no major complications. A written informed consent was obtained from the patient to publish this paper.

## 3. Discussion

SAM with LVOTO is a rare cardiac event that may occur during robotic thoracic surgery, which requires prompt diagnosis and tailored hemodynamic, ventilatory, and pain management to avoid perioperative shock and other serious hemodynamic complications. 

### 3.1. Hemodynamic Management Considerations

SAM induces an impediment in the LV outflow tract, leading to variable degrees of mitral valve regurgitation, reduction in stroke volume, and hypotension. Early detection of SAM requires the use intraoperative TEE. The hemodynamic management of SAM is based on three main pillars: increasing preload, decreasing heart rate, and increasing afterload (Figure 2). SAM increases as ventricular volume decreases, and it is relieved by volume expansion [11]. A large left ventricle end-diastolic volume, which reduces the mitral–septal contact, is a protective factor in preventing LVOTO. On the contrary, a small hyperdynamic and hypertrophic left ventricle represents a risk factor for SAM and LVOTO development. Fluid administration is the first-line therapy for preload reduction and acute hypotension. In addition, slowing the heart rate allows the heart to have enough time to fill and eject blood [12]. As a result, the maintenance of sinus rhythm and bradycardia preserves both atrial kick and forward stroke volume, reducing the risk of SAM and LVOTO development. Any tachyarrhythmia increasing SAM-related obstruction should be prevented and promptly treated [13]. Therefore, we suggest administering ultra-short-acting beta-blockers (e.g., esmolol or landiolol) [14] to decrease the heart rate and contractility. Indeed, long-acting beta-blockers are not appropriate in SAM management as hemodynamic parameters rapidly change. 

Diminished LV afterload increases SAM-related LVOTO. The increase in aortic pressure leads to higher LV end-diastolic pressure with the net effect of increasing the distance between the MV coaptation point and the septum, leaving LVOT open during the systole. This is the rationale for administering α-1 agonists (e.g., norepinephrine, phenylephrine, vasopressin) to increase the afterload [15] and to avoid drugs that diminish the peripheral resistances (e.g., nitroglycerin, α -2 agonist, nitroprusside, urapidil) [16]. Agents with selective tropism for the arterial tone should be preferred over nonselective agents. Therefore, etilefrine, which acts on β-1 and β-2 receptors, causes tachycardia and further worsens LVOTO. For the same reason, dopamine, dobutamine, and epinephrine should be avoided as they exert positive ionotropic and chronotropic effect [17]. In patients with SAM, hemodynamic instability can be exacerbated by hypovolemia (bleeding), vasodilation, and hyperdynamic ventricular function. Severe SAM-related LVOTO mimics cardiogenic shock that, if not promptly diagnosed, is treated with inotropes and diuretics. Indeed, these drugs worse LVOTO and mitral valve regurgitation. During induction of general anesthesia, careful titration of anesthetic drugs is required to reduce the risk of drug-induced hypotension and the activation of the sympathetic tone. During induction, it is not rare to observe desaturation unresponsive to the increase in the respiratory volumes. This can be explained by the drop in stroke volume, which occurs secondary to LVOTO. Furthermore, sevoflurane should be preferred over other volatile anesthetics to maintain anesthesia for its well-known mild myocardial depressant effect and a limited action on systemic vascular resistance and blood pressure [18,19]. During RATS, insufflation of CO_2_ can cause adverse effects on hemodynamics, which are amplified by the positive pressure ventilation. Jones et al. [20] observed a decrease in cardiac index, mean arterial pressure (MAP), and stroke volume and an increase in central venous pressure (CVP) with the effects proportional to the CO_2_ insufflation pressure [21,22,23,24,25]. In light of this, thoracic surgeons should relieve CO_2_ insufflation pressure in patients at higher risk of SAM or LVOTO. 

### 3.2. Respiratory Management Considerations

Heart–lung interaction during mechanical ventilation is well known. OLV is particularly challenging for RV. Indeed, the combination of the decrease in venous return secondary to high intrathoracic pressure and the increase in PAPs are associated with a drop in right ventricular output and consequent right ventricle failure. Positive pressure ventilation decreases venous return to the RV and increases pulmonary pressure, leading to a dilation of the RV, leftward shift of the interventricular septum, and limitation in LV filling (ventricular interdependence) (Figure 3). The addition of PEEP decreases LV afterload and reduces the gradient between LV and the aorta, eliciting SAM-related LVOTO. The addition of CO_2_ in RATS may exacerbate gas exchange impairment during OLV. Moreover, in patients undergoing RATS, there is an intrinsic risk of CO_2_ embolization and contralateral pneumothorax when the contralateral pleura is damaged. This risk should not be underestimated as it can influence LV preload, decreasing the venous return and increasing the venous pulmonary pressure, which in the end may elicit LVOTO in patients with predisposing factors for SAM development. 

Consequently, hypovolemic conditions, especially during mechanical ventilation, shift to the left the pressure–volume curve, increasing the risk of SAM, which is a preload-dependent phenomenon [26]. The alterations in transpulmonary and intrathoracic pressures that occur during tidal ventilation acting on left and right ventricular (RV) functions affect the right and left ventricle filling with a net effect of reducing the forward stroke volume. This constitutes the rationale for the use of small tidal volumes, also during the induction of general anesthesia, in patients at risk of SAM and LVOTO. Notably, as in the present case report, the desaturation observed during the induction of general anesthesia in patients with SAM and LVOTO is related to the drop in stroke volume rather than alveolar hypoventilation. Therefore, reduction of minute volume during induction, which is “per se” counterintuitive during desaturation, is the key to improve oxygenation, relieve SAM, and increase systemic stroke volume.

### 3.3. Pain Management Consideration

Analgesic techniques that mostly reduce systemic vascular resistances and induce vasodilation should be used with caution in patients with hypertrophic cardiomyopathy. The most effective technique for acute pain management after thoracic surgery is thoracic epidural analgesia (TEA). Several surveys have demonstrated that more than 50% (54%–85.6%) of thoracic anesthesiologists use TEA as the first-line treatment option for pain management [27,28,29,30,31,32]. However, TEA induces vasodilation below the level of the block, commonly resulting in hypotension, which may be exacerbated and have negative consequences for patients with higher risk of SAM and LVOTO [33]. In these circumstances, the paravertebral blockade is a valid alternative to TEA. Paravertebral blockade (PVB) reduces the risk of developing postoperative hypotension compared with thoracic epidural blockade with a similar effective profile in controlling acute pain [34,35]. Additionally, the use of opioids as analgesics leads to vasodilation and hypoventilation, which increase the risk of LVOTO. This aspect is particularly relevant in the management of ipsilateral shoulder pain (ISP), which commonly requires a high dose of opioids. The use of the interscalene block for pain control is a safe and effective alternative to intravenous analgesia to prevent LVOTO in patients at high risk of SAM. Therefore, in these patients, PVB and/or thoracic wall blocks should be preferred to TEA.

## 4. Conclusions

The management of patients with SAM in RATS includes peculiar treatment strategies, which, if not promptly and properly applied, may lead to catastrophic consequences. Notably, cases of SAM-related LVOTO are already reported in noncardiac surgery [36,37] and thoracic surgery [38] but never in RATS. In particular, the only case report in thoracic surgery was reported by Lasala et al., which observed the development of SAM in a patient with postural hypotension 2 days after surgery in the left thoracotomy. Since the intraoperative course was uneventful, the authors did not provide specific considerations to prevent and counteract SAM. 

On this basis, RATS still deserves specific considerations with respect to other noncardiac procedures. Thus, the present paper, which reports the first case of a patient with intraoperative SAM-related LVOTO during robotic lung major resection, may be useful in increasing knowledge in this field, highlighting the importance of being aware of this relatively neglected complication. The management of hemodynamic shock related to LVOTO and SAM requires a counterintuitive use of drugs and ventilation and a close cooperation between thoracic surgeons and anesthesiologists. Unfortunately, the current guidelines do not clearly recommend either the preoperative echocardiogram (TTE) or the intraoperative TEE for the assessment and management of patients with predisposing factors of SAM undergoing thoracic surgery. In fact, being SAM a dynamic condition, we strongly think that, at least in this highly selected population, TTE and TEE are complementary tools able to guarantee an uneventful course.

## Figures and Tables

**Figure 1 jcm-11-06044-f001:**
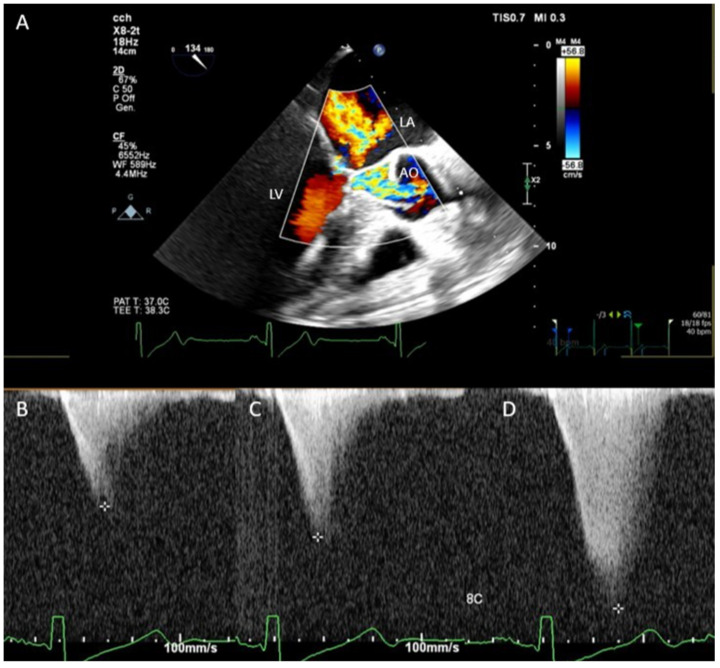
(**A**) Transesophageal color Doppler method showing the systolic anterior motion of the mitral valve and aliased flow in the left ventricular outflow tract during systole. Continuous wave Doppler showed a gradient ranging between 50 (**B**), 80 (**C**), and 180 mmHg (**D**) in different steps of the surgical procedure. LA, left atrium; LV, left ventricle; AO, aorta.

**Figure 2 jcm-11-06044-f002:**
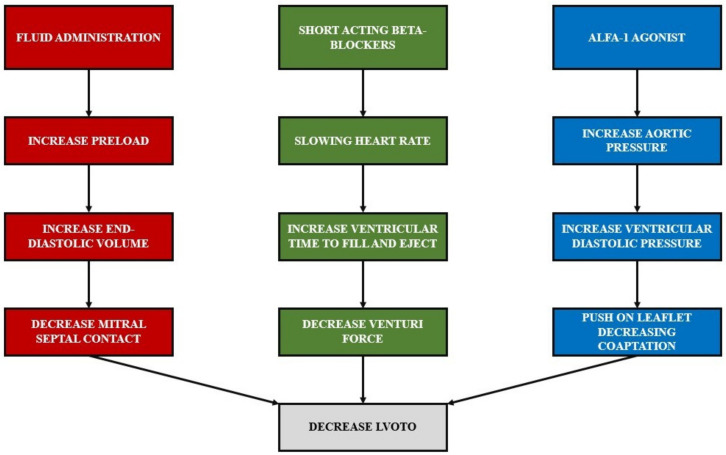
Three pillars of the hemodynamic management of SAM: increase preload, decrease heart rate, and increase afterload.

**Figure 3 jcm-11-06044-f003:**
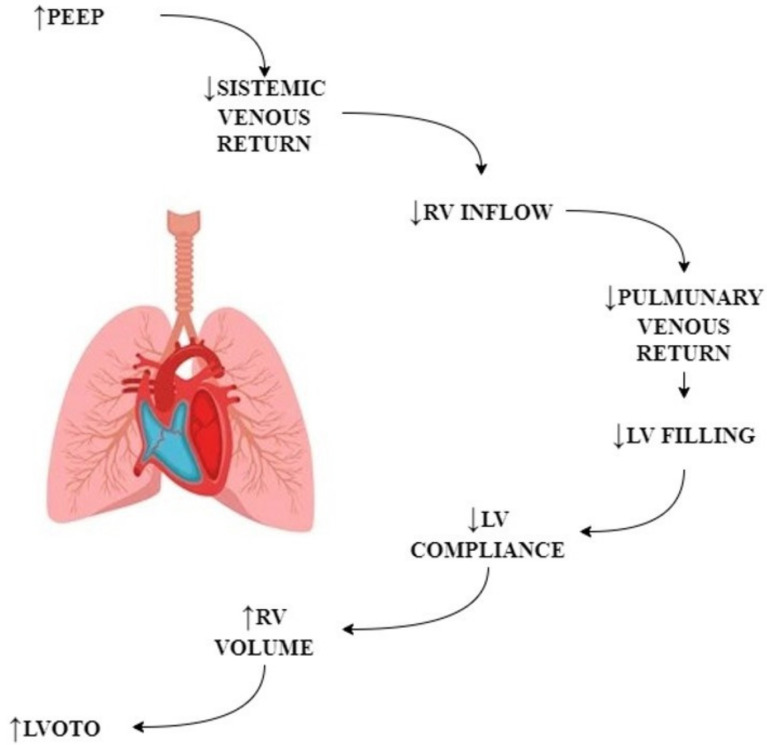
Hemodynamic consequences of mechanical ventilation.

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
