# Peer review of "Mitral Valve Systolic Anterior Motion in Robotic Thoracic Surgery as the Cause of Unexplained Hemodynamic Shock: From a Case Report to Recommendations"

_jcm, 2022, doi:10.3390/jcm11206044_

Round 1
Reviewer 1 Report
Dear Editor and Dr. Monaco and co-authors,
Thank you for asking me to review this work titled “Mitral Valve systolic anterior motion in robotic thoracic surgery as the cause of "covered" hemodynamic shock: from a case report to recommendations.” from San Raffaele Scientific Institute in Milan, Italy.
In this case report the authors wish to present their experience with hemodynamic instability caused by systolic anterior motion (SAM) of the mitral valve during robotic lobectomy. They present a case of a 70 year old patient and utilize this opportunity to provide a review of the management guidelines and suggestion.
This is an interesting case and I particularly liked the extensive review of the guidelines. I only have some minor suggestions for improvement. Thank you and good luck.
Comments:
1. Department is spelled wrong on the first affiliation.
2. The language of the abstract and at sections of the manuscript is very rough and it is difficult at times to understand what the authors are trying to say. It needs a major re-write by a native speaker or a professional editor!!!
3. If SAM was discovered immediately after anesthesia induction by TEE how was that related to the surgery which presumably had not commenced already?
4. Why was the SAM not recognized during the pre-operative cardiac work up of the patient? The patient had “hypertrophic ischemic cardiomyopathy NYHA class I” so why did the echo pre-op not reveal the SAM??
5. Why is “Robotic thoracic surgery under general anesthesia, mechanical ventilation, and one-lung ventilation (OLV) associated with bleeding, hypercapnia, and hypoxia is the "perfect storm" for the development of SAM and LVOTO”? One would assume this occurs in any type of major thoracic surgery with lung excision - why is robotic surgery different to open technique or for that matter VATS?
Reviewer 2 Report
Please see comments in attached SAM file.

Round 2
Reviewer 2 Report
Following revision and comments by the authors I am happy to recommend publication.